# Conservation of dark CPD photolyase function in blind cavefish

Hongxiang Li[1], Carina Scheitle[1], Giuseppe Di Mauro[2], Silvia Fuselli[3], Susanne Fritsch-Decker[1], Takeshi Todo [4], Carsten Weiss[1], Daniela Vallone[1], Tilman Lamparter [5], Cristiano Bertolucci [3] & Nicholas S. Foulkes [1] ✉

DNA damage is generated by various environmental stressors and so DNA repair systems must inevitably adapt to changing environments. Photolyases represent a highly conserved class of enzymes which repair UV-induced covalent crosslinks between adjacent pyrimidine bases (CPD and 6-4 photo-products) via photoreactivation. In the blind cavefish *Phreatichthys andruzzii* which has evolved for millions of years completely isolated from UV radiation and visible light, we have documented multiple polymorphisms and loss of function mutations affecting both the 6-4phr and DASHphr photolyase genes while strangely, the CPDphr gene remains highly conserved. Using loss and gain of photolyase function medaka and mammalian cell lines, we reveal a novel function for CPDphr. Specifically, it enables the light-independent repair of CPD as well as 8-OHdG, an oxidatively modified form of guanosine which are both generated under oxidative stress in the absence of UV radiation. Thereby we document selective conservation of light-independent photolyase function in blind cavefish, enabling the repair of DNA damage encountered in an extreme subterranean environment.

Sunlight represents a critically important environmental factor for supporting life on earth. Over the course of evolution, while cells and organisms have developed various mechanisms to perceive sunlight and to effectively mobilize it as an energy source, it is also an environmental stressor. Exposure to UV radiation in sunlight can lead to the formation of carcinogenic DNA lesions[1,2] including cyclobutane pyrimidine dimers (CPDs) and (6-4) photoproducts of pyrimidine dimers (6-4PPs). One of the most conserved mechanisms for preserving genome integrity upon exposure to sunlight is photoreactivation. This mechanism harnesses visible light as a source of energy to repair UV-damaged DNA and is encountered in most living systems from bacteria to vertebrates[3]. Photoreactivation is catalysed by a highly conserved class of flavoproteins, termed photolyases[3,4]. These enzymes employ flavin and other cofactors to serve as chromophores for enabling photorepair. Three main categories of photolyases have been

identified. CPD photolyase (CPDphr) and Cry-DASH photolyase (DASHphr) repair CPDs in the context of double-stranded and single-stranded DNA, respectively, while 6-4 photolyase (6-4phr) repairs 6-4PPs[5–7]. Strangely, placental mammals do not possess photolyase genes. Instead, they use the more versatile, yet less efficient Nucleotide Excision Repair (NER) to repair UV-induced DNA damage. Another notable example of the loss of photoreactivation is in the blind Somalian cavefish (*Phreatichthys andruzzii*), a species completely isolated from sunlight for millions of years in phreatic layers deep beneath the Somalian desert[8]. Evolution in a perpetually dark environment may well account for the loss of photoreactivation in this species[9]. Indeed, changes in the DNA repair repertoire of this cavefish over the course of evolution may be regarded as a response to the different types of DNA damage typically encountered in subterranean environments. For example, life in hypoxic conditions generates

[1]Institute of Biological and Chemical Systems - Biological Information Processing, Karlsruhe Institute of Technology, Eggenstein-Leopoldshafen, Germany. [2]International Centre for Genetic Engineering and Biotechnology, Trieste, Italy. [3]Department of Life Science and Biotechnology, University of Ferrara, Ferrara, Italy. [4]Radioisotope Research Center, Institute for Radiation Science, Osaka University, Osaka, Japan. [5]Botanical Institute, Karlsruhe Institute of Technology, Karlsruhe, Germany. ✉e-mail: nicholas.foulkes@kit.edu

oxidative stress, which can lead to a significant increase in certain types of DNA damage[10,11]. Therefore, a study of the evolution of DNA repair systems potentially provides important insight into their functionality as well as the evolutionary pressures that shape them.

Based on a pattern of selective loss of photolyase gene function in the cavefish *P. andruzzii*, we reveal an important light-independent function for CPD photolyase. We demonstrate that while the gene pool of our *P. andruzzii* colony contains multiple loss-of-function alleles for 6-4phr, the CPDphr gene is conserved. To explore CPDphr function in more detail, we examined DNA repair in a set of CRISPR-generated, loss-of-photolyase-function medaka lines. This revealed that as well as attenuation of photoreactivation repair in all three lines, the CPDphr mutants, but not the 6-4phr or DASHphr mutants exhibited increased susceptibility to DNA damage upon elevated oxidative stress. Furthermore, ectopic expression of zebrafish CPDphr in mouse cell lines conferred not only photoreactivation but also enhanced their resistance to oxidative stress in a light-independent manner. Loss of CPDphr in medaka is associated with increased levels of CPDs as well as 8-hydroxydeoxyguanosine (8-OHdG or 8-oxodG) damage, an oxidatively modified form of guanosine, even under constant darkness in vivo and in vitro. Furthermore, CPD photolyase is able to bind to DNA incorporating 8-OHdG as well as CPDs. The protective effect of ectopic CPD photolyase expression in mammalian cells is dependent upon the integrity of a set of highly conserved tryptophan residues within the active site of the protein, which are also critical for mediating photoreactivation.

These results point to CPDphr being involved in the light-independent repair of CPDs as well as 8-OHdG and thereby are consistent with the hypothesis that its functional conservation during evolution in an extreme subterranean environment has enhanced the repair of oxidatively damaged DNA.

## Results

### Conservation of CPDphr, but not the 6-4phr gene in blind cavefish

In a study of DNA repair systems in the Somalian cavefish, *P. andruzzii*, we have previously documented several loss-of-function mutations affecting photolyase genes based on RTPCR amplification of cDNA prepared from RNA extracts of pools of embryos[9]. In order to explore the photolyase genotype at the individual animal level, we prepared DNA extracts from fin clips collected from 6 individual adult fish in our colony. This colony was established from more than 100 original specimens collected at Bud Bud (Galguduud region), in Somalia in 1982 and currently consists of mainly F0, F1 and F2 generation fish which are expected to represent the original population in terms of genetic variation. The results from PCR amplification followed by sequence analysis confirmed the presence of multiple loss-of-function mutations affecting 6-4phr (Fig. 1). These predicted the presence of point mutations that generate C-terminally truncated proteins. These mutant 6-4phr proteins are unable to enter the nucleus or access damaged DNA[9]. In contrast, the CPDphr genes in all these individuals appeared highly conserved, showing no polymorphic sites within *P. andruzzii*. These results are consistent with differential conservation of the different members of the photolyase gene family and suggest that retaining normal CPDphr function could provide a potential benefit in environments that lack exposure to visible or UV light.

### Loss of CPDphr function in medaka results in increased DNA damage and cell mortality upon oxidative stress

One commonly encountered consequence for life in subterranean and cave environments is an increased level of oxidative stress, which can originate from exposure to hypoxic conditions or metabolic stress related to reduced nutrient availability[11]. These stressors can serve as an important source of DNA damage. So, might CPDphr play some role in DNA repair and survival under oxidative stress? To explore in more

detail the functions of CPD, 6-4 and DASH photolyase, we wished to study the phenotypes associated with their loss of function in a genetically tractable, epigean fish species. As well as offering a comprehensive set of genetic tools including CRISPR-Cas9 gene editing, medaka wildtype lines such as iCab have tolerated close to 100 generations of brother-sister crosses and so can be considered effectively isogenic. We therefore employed CRISPR-Cas9-generated medaka lines carrying loss-of-function mutations in each of the three photolyase genes, their corresponding wild type (WT) control lines as well as primary cell lines prepared from each fish line[12]. All these lines share the same iCab isogenic background and so phenotypic differences observed in these mutants are more likely related to the photolyase mutations and not the result of linkage with other polymorphic genes. Previously, a deficiency in photoreactivation of UV-induced DNA damage has been documented in these mutant lines[12]. We next tested the effect of increased oxidative stress on DNA repair activity in the various cell lines. Specifically, all the photolyase mutant cell lines and their WT controls were exposed to different concentrations of $H_2O_2$ and were maintained under constant darkness (DD) to avoid photoreactivation. We then measured endogenous levels of phosphorylated Histone H2AX at Ser 139 ($\gamma$-H2AX)[13–15], which are widely considered as a proxy for levels of DNA damage. We observed a notable oxidative stress-induced increase in $\gamma$-H2AX levels in the CPDphr mutant cell line compared with WT controls, whereas no significant differences between WT and loss-of-function DASHphr and 6-4phr cells were detected (Fig. 2a–c; Supplementary Fig. 1). Might this differential DNA damage response be limited to these cell culture models? Therefore, to confirm these initial in vitro results, we tested the endogenous levels of oxidative stress-induced $\gamma$-H2AX in explanted cultures of fin clips from the photolyase mutant fish[12] as well as their corresponding WT controls following challenge with $H_2O_2$. Consistent with the in vitro results, we observed a significant increase in $H_2O_2$-induced $\gamma$-H2AX levels in the CPDphr mutant compared with WT fin clips. However, this was not observed in 6-4phr and DASHphr mutants (Fig. 2d; Supplementary Fig. 2). These results point to loss of CPDphr, but not 6-4phr and DASHphr, resulting in a rise in DNA damage induced by oxidative stress.

Increased levels of DNA damage are typically associated with a reduction in cell viability. Therefore, to confirm our initial results based on $\gamma$-H2AX levels, we next focused on the CPDphr mutant cell line and its WT control and assayed cell viability by performing an automated high-throughput microscopy (AHM) assay. Following $H_2O_2$ treatment under constant darkness, the cells were stained with Hoechst and Propidium Iodide (PI) and then examined by microscopy. Differential staining intensity with these two dyes reveals different cell populations, categorized as living, early and late apoptotic and necrotic cells. Compared with the medaka WT cells, the CPDphr mutant cells showed reduced cell survival upon elevated oxidative stress (Fig. 2e and Supplementary Fig. 3). In order to confirm the AHM assay results, another approach of staining cells with 3-(4,5-dimethyl-thiazol-2-yl)-2,5-diphenyltetrazoliumbromid (MTT), was used to assess cell viability. In the mitochondria of living cells, MTT is reduced to purple formazan by mitochondrial dehydrogenase and the amount of formazan is proportional to the number of living cells. This confirmed that the cell mortality of CPDphr-mutant cells was significantly increased following exposure to elevated $H_2O_2$ levels under constant darkness, compared with the WT cells (Fig. 2f).

**Gain of CPDphr function confers enhanced cell survival and reduced levels of DNA damage upon oxidative stress in mammalian cells.** Placental mammals lack photolyase genes and photoreactivation. Therefore, would ectopic expression of fish CPDphr in a mammalian cell line confer photoreactivation as well as enhanced resistance to oxidative stress-induced DNA damage? Furthermore, given the high degree of conservation of the CPDphr gene

**a**

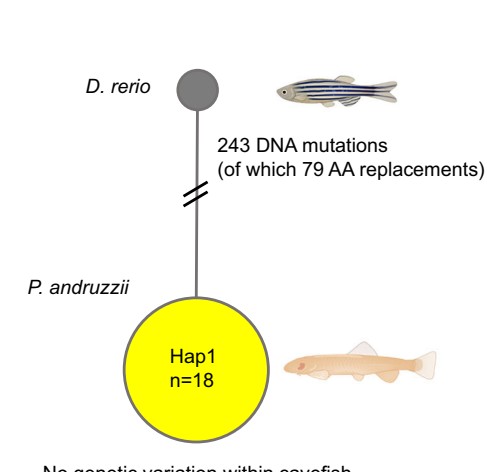

**b**

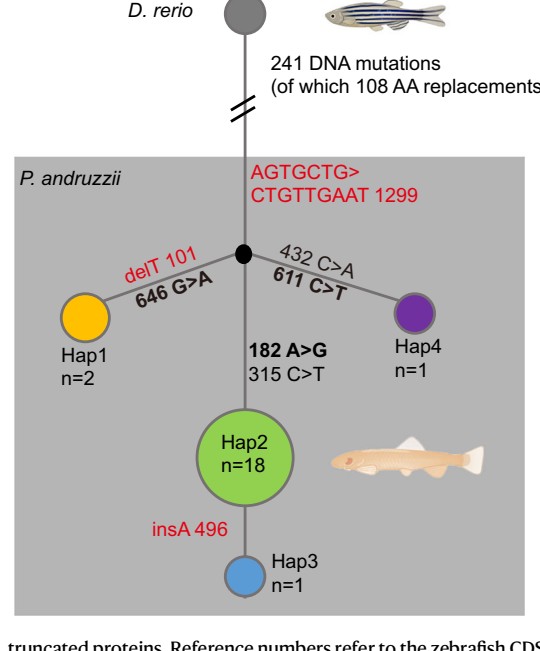

**Fig. 1 | CPDphr but not 6-4phr is conserved in blind cavefish.** Summary of genome sequencing analysis performed in zebrafish (*D. rerio*) and cavefish (*P. andruzzii*) with the CPDphr (**a**) and 6-4phr (**b**) haplotypes represented in a network. The size of the circles is proportional to the haplotype frequency in the dataset. Coding-region mutations separating haplotypes are marked in the figure and indicated by their position in the coding sequence (CDS). Mutations causing premature stop codons are indicated in red while nonsynonymous mutations are indicated in bold. All 6-4phr haplotypes in the shaded area encode for C-terminally truncated proteins. Reference numbers refer to the zebrafish CDS (dark-grey circle; accession numbers: CPDphr NM 201064.1; 6-4phr NM_131788.1). Total number of sites analyzed: CPDphr 1476 bp; 6-4phr 1562 bp. AA amino acids. The origin of the individual *P. andruzzii* fish used in this sequence analysis is summarized in Supplementary Table 1. The CPDphr and 6-4phr sequences have been submitted to GenBank (Accession numbers for *P. andruzzii* CPDphr (PV951863) and 6-4phr (PV983310, PV983311, PV983312 and PV983313)). Created in BioRender. Foulkes, N. (2025) https://BioRender.com/1osqe99.

during evolution of non-mammalian vertebrates[16,17], do CPDphrs from fish species other than medaka also enhance the protective response to oxidative stress–induced DNA damage? To address these questions, we stably transfected mouse 3T3 fibroblast cells with an expression vector for zebrafish CPDphr and initially confirmed ectopic expression of the fish-derived transgene in the resulting cell line (3T3 CPD) by western blotting (Supplementary Fig. 4a). As a positive control we then tested whether the ectopically expressed CPDphr confers photoreactivation on 3T3 cells. Specifically, after transient exposure to various doses of UV-C light, the 3T3 CPD cell line as well as non-transfected 3T3 cells were transferred to DD or light-dark (LD) cycle conditions. The viability of 3T3 CPD cells was substantially higher after recovery under LD conditions compared to cells maintained in darkness, while there was no discernible influence of light on survival in non-transfected 3T3 cells, thus confirming that ectopic-expression of zebrafish CPDphr in mouse cells is sufficient to confer photoreactivation (Supplementary Fig. 4b, c).

Our next step was to investigate how the response to oxidative stress was affected in 3T3 CPD cells. This cell line together with non-transfected 3T3 cells was transiently exposed to increasing doses of $H_2O_2$ and then transferred to DD. Both cell lines showed a dose-dependent increase in γ-H2AX levels, while 3T3 CPD cells exhibited consistently reduced levels of γ-H2AX staining compared to non-transfected 3T3 cells at all $H_2O_2$ doses tested (Fig. 3a, Supplementary Fig. 5). These results illustrate that ectopic expression of CPDphr reduces DNA damage upon challenge with oxidative stress. We confirmed these findings by comparing the survival of non-transfected 3T3 and 3T3 CPD cells upon $H_2O_2$ treatment using AHM as well as the MTT staining assay. Exposure of 3T3 cells to increasing levels of $H_2O_2$ led to a dose-dependent decrease in cell viability. However, consistent with

the γ-H2AX results, ectopic expression of zebrafish CPDphr enhanced cell viability across a range of $H_2O_2$ concentrations (Fig. 3b, c and Supplementary Fig. 6). Therefore, ectopic CPDphr expression confers reduced DNA damage and increased resistance to oxidative stress.

**Potential targets for CPDphr function in darkness**
Specifically, which types of DNA damage might CPDphr target in fish isolated from sunlight in a perpetually dark subterranean environment where presumably, UV-induced CPD is absent? To address this question, we measured changes in the levels of specific types of DNA damage in our loss-of-CPDphr-function medaka line. Elevated CPD levels induced by UV exposure have already been documented in these mutant cells[12]. Interestingly, it has been reported that CPD continues to be generated in complete darkness, several hours following UV exposure[18], the so-called "dark" CPD. Initially, we quantified basal CPD levels in genomic DNA prepared from the brain and liver of wild-type iCab medaka and CPDphr mutant fish by ELISA assay. These two organs were selected since they represent high metabolic rate tissues and therefore might also tend to experience elevated ROS levels[19]. Prior to this analysis, the fish were maintained under standard laboratory husbandry lighting conditions in the complete absence of UV radiation. Interestingly, compared with iCab, basal levels of CPD in the liver of CPDphr-mutated fish were significantly elevated, whereas CPD levels in the brain of the mutants were comparable with those of iCab fish (Fig. 4a, b). Given that previous studies have implicated redox reactions in the generation of CPD[18], we wondered whether CPD may be generated in the CPDphr mutant tissues as the consequence of metabolic activity. Specifically, might elevated ROS levels represent a source of CPD? To address these questions, we initially exposed the medaka WT and CPDphr mutant cell lines to $H_2O_2$ under constant

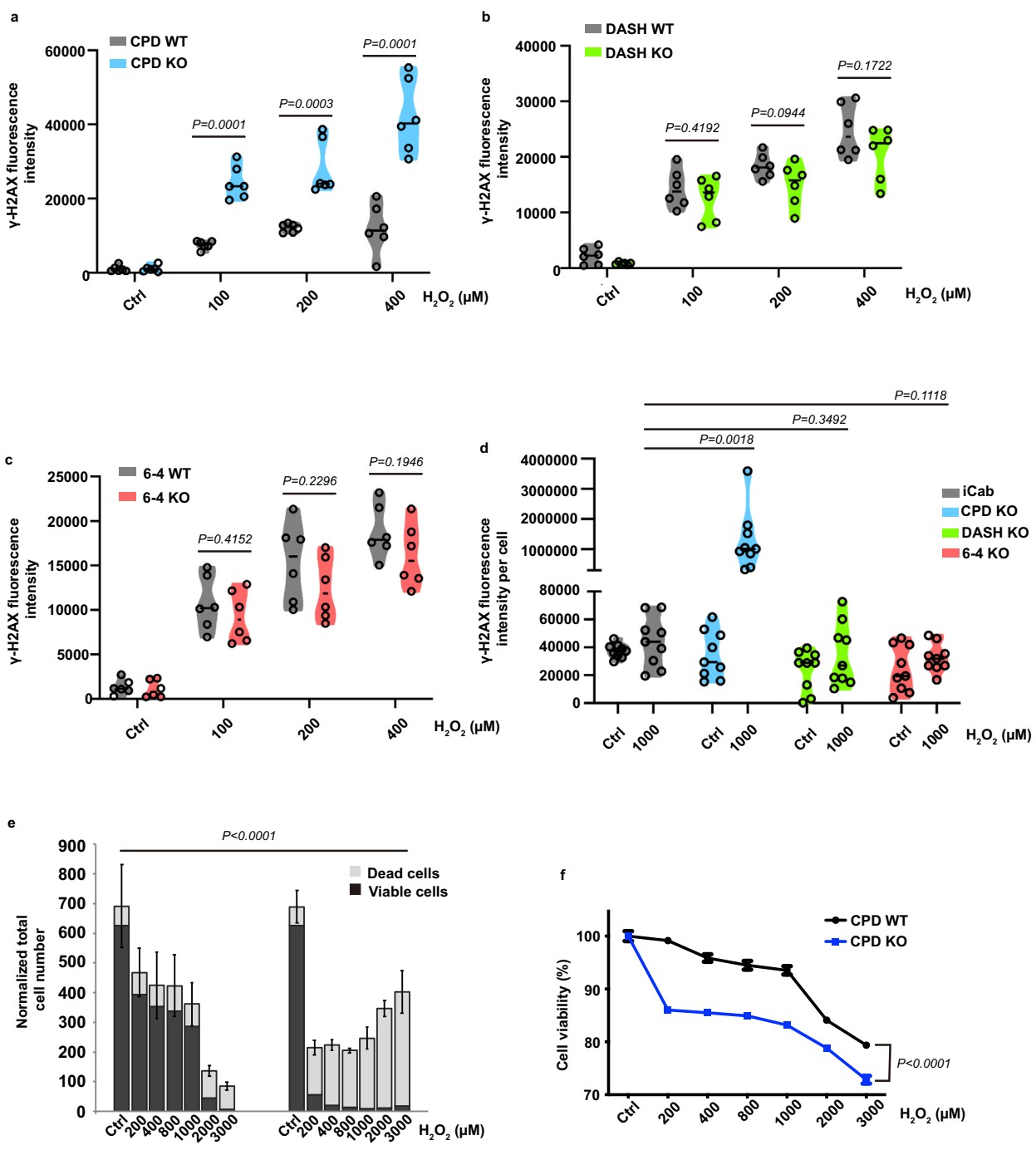

**Fig. 2 | Loss of CPDphr function in medaka results in increased DNA damage and cell mortality upon oxidative stress.** Immunofluorescence assays measuring γ-H2AX levels in medaka wild type (grey truncated violin (WT)) as well as CPD (blue truncated violin (**a**)), DASH (green truncated violin (**b**)) and 6-4 (red truncated violin (**c**)) photolyase mutant (KO) cells treated with various concentrations of H2O2 as indicated on the x-axes. **d** Immunofluorescence assay of γ-H2AX levels in wild-type medaka (iCab) and photolyase mutant (KO) fish fin clips treated with 1000 μM H2O2 or untreated controls (Ctrl) as indicated on the x-axis. Violin plots represent individual data points (**a**–**c**, $n = 6$ biologically independent samples; **d**, $n = 9$ biologically independent samples) shown as black hollow circles and their probable distribution. The horizontal black lines within the violin plots represent mean values of γ-H2AX fluorescence intensity. **e** Quantification of automated high-throughput microscopy (AHM) assay results in medaka CPD WT and KO cells exposed to a range of H2O2 concentrations up to 3000 μM. Total numbers of dead and viable cells are plotted as means ± SEM ($n = 4$ biologically independent

samples), normalized in relation to untreated cells (Ctrl) on the y-axis, while H2O2 concentrations are indicated on the x-axis. Note that the overall reduced cell counts in the right panel (CPD KO) compared with the left panel (CPD WT) may be a consequence of reduced cell proliferation and higher levels of cell death in the mutant cells resulting in their detachment from the cell culture surface. **f** Cell viability assay of medaka CPD WT and KO cells exposed to a range of H2O2 concentrations from 200 to 3000 μM. Mean percentage ± SEM ($n = 8$ biologically independent samples) of cell viability with respect to untreated cells are plotted on the y-axis, while H2O2 concentrations are indicated on the x-axis. All assays were performed in constant darkness (DD) and repeated at least 3 times, independently. For the cell viability assays (**e**, **f**), representative data is shown. The statistical test used for (**a**–**d**) is student's t-test, while the test for (**e**, **f**) is two-way analysis of variance (ANOVA) test. Statistical differences (P values) are indicated in each panel. Source data are provided as a Source Data file.

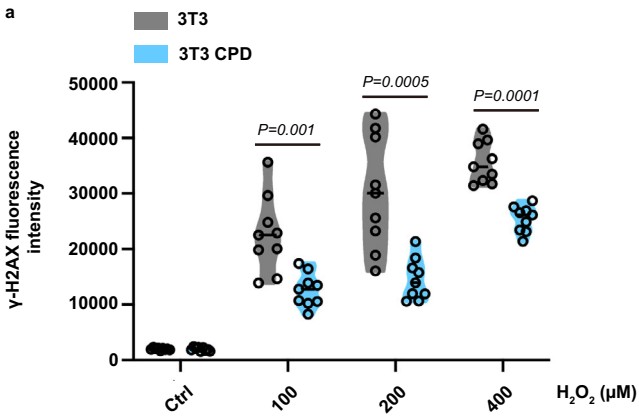

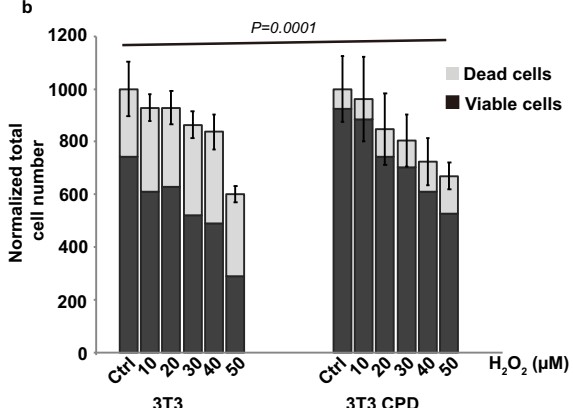

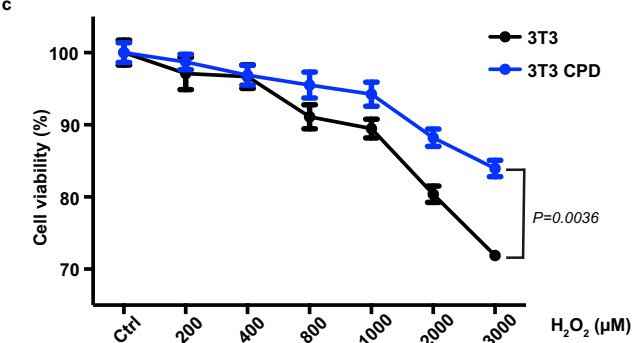

**Fig. 3 | Gain of CPDphr function confers enhanced cell survival and reduced levels of DNA damage upon oxidative stress in mammalian cells. a** Violin plot showing quantification of immunofluorescence assay measurements of γ-H2AX levels in 3T3 (grey) and 3T3 CPD cells (blue) treated with a range of $H_2O_2$ concentrations up to 400 μM indicated on the x-axis. The horizontal black lines within the violin plots indicate mean values and the black hollow circles represent individual data points ($n = 9$ biologically independent samples). **b** AHM assay results in 3T3 and 3T3 CPD cells exposed to a range of $H_2O_2$ concentrations up to 50 μM. Total numbers of dead and viable cells are plotted as means ± SEM ($n = 4$ biologically independent samples) normalized in relation to untreated cells (Ctrl) on the y-axis, while $H_2O_2$ concentrations are indicated on the x-axis. **c** MTT assay results of 3T3 and 3T3 CPD cells exposed to $H_2O_2$ concentrations ranging from 200 to 3000 μM. Mean percentage ± SEM ($n = 8$ biologically independent samples) cell viability with respect to untreated cells is plotted on the y-axis, while $H_2O_2$ concentrations are indicated on the x-axis. Controls confirming that the transfection and subsequent selection procedures are not responsible for the observed results with the 3T3 cells are provided where mutant forms of CPDphr were ectopically expressed in 3T3 cells and results equivalent to those from untransfected 3T3 cells were obtained (see below). All experiments were conducted in constant darkness (DD) and repeated at least 3 times, independently. For the cell viability assays (**b, c**), representative data is shown. The statistical test used for (**a**) is student's t-test, while the analysis for (**b, c**) is two-way ANOVA analysis. Statistical differences (P values) are indicated in each panel. Source data are provided as a Source Data file.

darkness and quantified CPD levels. Surprisingly, we observed a substantial increase in the levels of CPD in the CPDphr-mutant cells while no significant changes in CPD was observed in the WT cells (Fig. 4c). Consistently, treatment of explant cultures of dissected fin clip tissue from iCab and CPDphr mutants with $H_2O_2$ under constant darkness resulted in increased levels of CPD in the CPDphr mutants (Fig. 4d). These results are consistent with elevated oxidative stress inducing CPD and that loss of CPDphr function results in accumulation of CPD. Interestingly, in the WT cells and fin clips, there is no evidence for ROS-induced CPD damage, even under constant darkness, suggesting that this damage might be repaired by CPDphr via a light-independent mechanism.

In addition to CPD, there are many other types of DNA damage generated by oxidative stress. One well studied example is 8-OHdG. We therefore initially quantified the basal levels of 8-OHdG in the brain and liver of medaka iCab and CPDphr mutant fish. In both tissues of the mutants, we observed elevated basal levels of 8-OHdG compared with iCab fish (Fig. 4e, f). Consistent with these in vivo results, the loss of CPDphr function medaka cell line showed higher basal and oxidative stress-induced 8-OHdG levels (Fig. 4g, h). Furthermore, $H_2O_2$ treatment of explanted fin clip cultures prepared from mutant fish resulted in

significantly elevated levels of 8-OHdG compared with iCab-derived fin clip cultures (Figs. 4g, i). Thus, our results point to CPDphr enhancing the repair of ROS-induced 8-OHdG and CPD under constant darkness.

While it is well documented that CPDphr binds with high affinity to CPD, can it also interact with 8-OHdG damaged DNA? To test this, we initially prepared zebrafish CPDphr protein in the Er2566 E. coli bacterial expression system, purified it by Ni affinity chromatography (Supplementary Fig. 7a) and then compared its binding to damaged and undamaged DNA by EMSA assay. We specifically tested binding of the purified CPDphr protein to a double-stranded oligonucleotide probe carrying 8-OHdG damage together with an undamaged oligonucleotide and a UV-irradiated probe carrying CPD modification as negative and positive controls, respectively. All probes shared the same nucleotide sequence apart from the differences at the single modification site. Our results revealed besides CPD, CPDphr protein also has the capacity to bind to 8-OHdG damage (Fig. 4j and Supplementary Fig. 7b, c).

## Light-independent DNA repair by CPD photolyase

We next explored the mechanism whereby CPD photolyase enhances the repair of oxidative stress-induced DNA damage. One of the

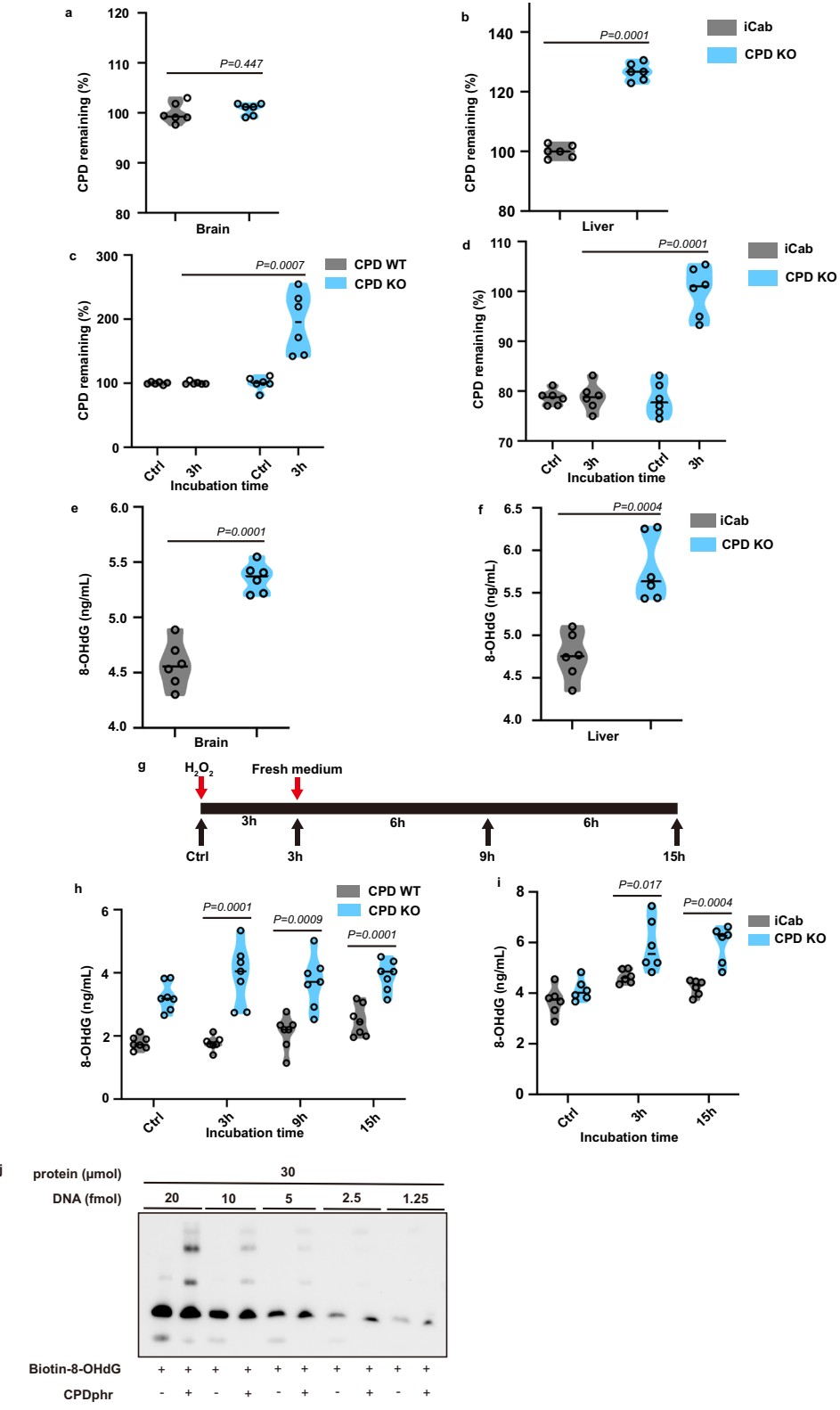

hallmarks of photolyase function in photoreactivation is that it is intricately connected with light exposure. These proteins absorb and transfer light energy via a flavin cofactor as well as an antenna chromophore and thereby specifically catalyse photoreactivation repair of CPD damage[3]. We therefore initially questioned whether light exposure might influence the repair of oxidatively damaged DNA by CPDphr?

We compared γ-H2AX levels in medaka WT cells following exposure to $H_2O_2$ under DD or LD conditions. Cells maintained under DD or LD without $H_2O_2$ exposure were used as controls. Following transient treatment with $H_2O_2$, γ-H2AX levels progressively increased during the subsequent recovery period. However, there were no significant differences in γ-H2AX levels between the cells exposed to DD and LD (Fig. 5a and Supplementary Fig. 8a, b). Consistently, no significant

**Fig. 4 | The potential targets of CPDphr function in darkness.** ELISA (enzyme-linked immunosorbent assay) analysis of basal levels of CPD in the brain (**a**) and liver (**b**) of medaka wild type (iCab) and CPD photolyase mutant fish (CPD KO). ELISA assay of CPD levels in medaka CPD WT and KO cells (**c**) and fin clips from iCab and CPD KO medaka (**d**) treated with 1000 μM $H_2O_2$ for 3 h. ELISA of basal levels of 8-hydroxydeoxyguanosine (8-OHdG) in the brain (**e**) and liver (**f**) of iCab and CPD KO medaka. **g** Experimental design for the analysis reported in (**h**, **i**). ELISA assay of 8-OHdG levels in CPD WT and KO cells (**h**) and fin clips from iCab and CPD KO medaka (**i**) treated with 1000 μM $H_2O_2$ for 3 h and then left to recover in constant darkness. The sampling timepoints are indicated on the *x*-axes. **a–f**, **h–i** Violin plots represent individual data points (**a**, **b** and **e**, **f**, *n* = 6 biologically independent samples with each sample including 9 fish. **c** *n* = 6 biologically independent samples. **d** *n* = 6 biologically independent samples with each sample including 35 fish. **h** *n* = 7 biologically independent samples. **i** *n* = 6 biologically independent samples with each sample including 49 fish) shown as black hollow circles and their probable distribution. The horizontal black lines within the violin plots represent mean values of CPD photoproduct (**a–d**) and of 8-OHdG (**e**, **f**, **h**, **i**). **j** EMSA (electrophoretic mobility shift assay) of bacterially synthesized and purified zebrafish CPDphr protein with biotin-labelled, titrated and commercially synthesized, 8-OHdG modified oligo. Quantities of protein (μmol) and oligonucleotide probe (DNA, fmol) included in each assay are indicated above each lane. All assays were repeated independently at least 2 times. The statistical test used for (**a–f**, **h**, **i**) is student's *t*-test. Statistical differences (*P* values) are annotated in each panel. Source data are provided as a Source Data file.

difference in cell viability was observed between the WT medaka cells recovering under DD and LD conditions following exposure to oxidative stress (Fig. 5b). Does light affect the survival of mammalian cells ectopically expressing CPDphr following exposure to $H_2O_2$? 3T3 CPD together with WT 3T3 cells were transiently exposed to various doses of $H_2O_2$ and allowed to recover in DD or LD conditions before assaying cell viability using the AHM assay or by MTT staining. The results demonstrate that cell survival decreased in a $H_2O_2$ dose-dependent manner, and that light exposure had no significant effect on survival in the 3T3 and 3T3 CPD cell lines (Fig. 5c–f and Supplementary Fig. 9). Together, these results point to oxidative stress-induced DNA damage repair by CPDphr being mediated by a light-independent mechanism.

The enzymatic mechanism whereby CPDphr catalyses photoreactivation has been extensively studied[20–23]. Within the 3D structure of the CPDphr protein, a triad of highly conserved tryptophan residues proximal to the active site is critical for electron transfer from the protein surface to the FAD cofactor[21–25]. Is the integrity of this three-tryptophan electron transfer chain essential for how CPDphr enhances DNA repair and survival upon oxidative stress under complete darkness? To tackle this question, we generated mutant versions of zebrafish CPDphr where two of the conserved triad of tryptophan residues were mutated to phenylalanine (W310F and W400F). Expression vectors for both mutants were then stably transfected into 3T3 cells and ectopic expression of the epitope tagged mutant proteins, as well as the WT control, were confirmed by western blotting (Supplementary Fig. 10a).

As a positive control, we initially tested the effect of loss of function of the tryptophan residues upon photoreactivation repair. We observed that compared with 3T3 CPD cells (Supplementary Fig. 4c), cells expressing the tryptophan mutants did not exhibit any significant increase in cell survival when recovering under LD conditions following UV exposure, consistent with the lack of photoreactivation in the mutant cell lines (Supplementary Fig. 10b, c). We subsequently examined whether the ectopic expression of the tryptophan mutant CPDphr affected the viability of mammalian cells when exposed to oxidative stress. The 3T3 cell lines were exposed to a range of doses of $H_2O_2$ and then allowed to recover in fresh medium under darkness. The results demonstrated that, compared with 3T3 CPD cells, the 3T3 CPD mutants showed reduced viability indicating that the integrity of the three-tryptophan electron transfer chain in CPDphr, is essential for both the light-dependent and light-independent repair functions in CPDphr (Fig. 5g).

## Discussion

The light-dependent enzymatic repair of pyrimidine dimers by photolyase has been studied extensively at the atomic, molecular and cellular levels but less at the evolutionary level. Differential functionality of the CPD, 6-4 and DASH photolyases has been attributed to each photolyase targeting different types of UV-induced damage (CPDs or 6-4PPs), within different DNA structures (double or single stranded)[5–7]. Here, by studying the evolution of photolyase genes in an extreme,

perpetually dark subterranean environment and by a comparative study using the zebrafish and medaka genetic models, with in vitro and in vivo analysis, we have revealed a novel, light-independent function for CPDphr in the repair of oxidative stress-induced DNA damage. Our observations indicate that CPDphr contributes to the repair of both 8-OHdG and CPD in the absence of visible light and UV irradiation. This involves CPDphr binding to CPD and 8-OHdG damage and the repair requires the integrity of the three-tryptophan Electron Transfer Chain at the active site of the protein.

Our observations that CPD levels are elevated in CPDphr mutant cells upon $H_2O_2$ treatment in the absence of UV light, are reminiscent of the previous reports of "dark" CPD that persists for several hours after UV exposure has ended[18]. It has been shown previously that in hairless albino mouse epidermis, the formation of CPD is notably higher 2–4 h after UV exposure in darkness compared to the levels immediately after UV light exposure[26]. Furthermore, in human unstimulated lymphocytes, the CPD levels reach a peak 4 h after being exposed to UV radiation[27]. The current models explaining dark CPD generation include redox changes occurring in the context of melanin metabolism. It has been well documented that exposure to UV radiation has a direct effect on adjacent pyrimidine bases in DNA, causing them to form covalent cross links. However, UV also represents a potent source of various types of ROS which can accumulate following the UV exposure and then together with melanin breakdown products cause the generation of CPD damage. However, dark CPD formation has also been seen in cell types that do not metabolize melanin, therefore indicating that this model does not provide a complete explanation for how this damage can be formed without sunlight. Our results go one step further and implicate elevated levels of ROS alone as being sufficient to drive CPD formation in darkness.

Our results have documented a differential impact of loss of CPDphr function on the relative basal levels of CPD and 8-OHdG in the medaka liver and brain. Many lines of evidence have implicated cell type-specific differences in the activities of DNA repair systems[28–30]. In addition, the response to DNA damage varies significantly over the course of the cell cycle as well as in tissue-specific stem cells[31,32]. Combined with fundamental differences in the oxidative stress response of hepatocytes and neurons, this may account for the differential accumulation of DNA damage observed in these two tissues[33,34].

An important question that is raised by these findings is how CPDphr is able to interact with 8-OHdG, given that the structure of this modified guanine base is very different from the pyrimidine dimers that constitute CPD. Interestingly, it has been previously reported that *E. coli* DNA photolyase can bind specifically to cisplatin and stimulate the removal of this lesion by excision nuclease, thereby enhancing resistance to cisplatin-induced cell death[35]. Moreover, yeast photolyase can bind to other types of DNA damage, although with lower affinity[36]. It seems unlikely that the interaction with these structurally distinct types of DNA damage could occur via the same active site of the photolyase protein, given the intricate 3D points of contact that

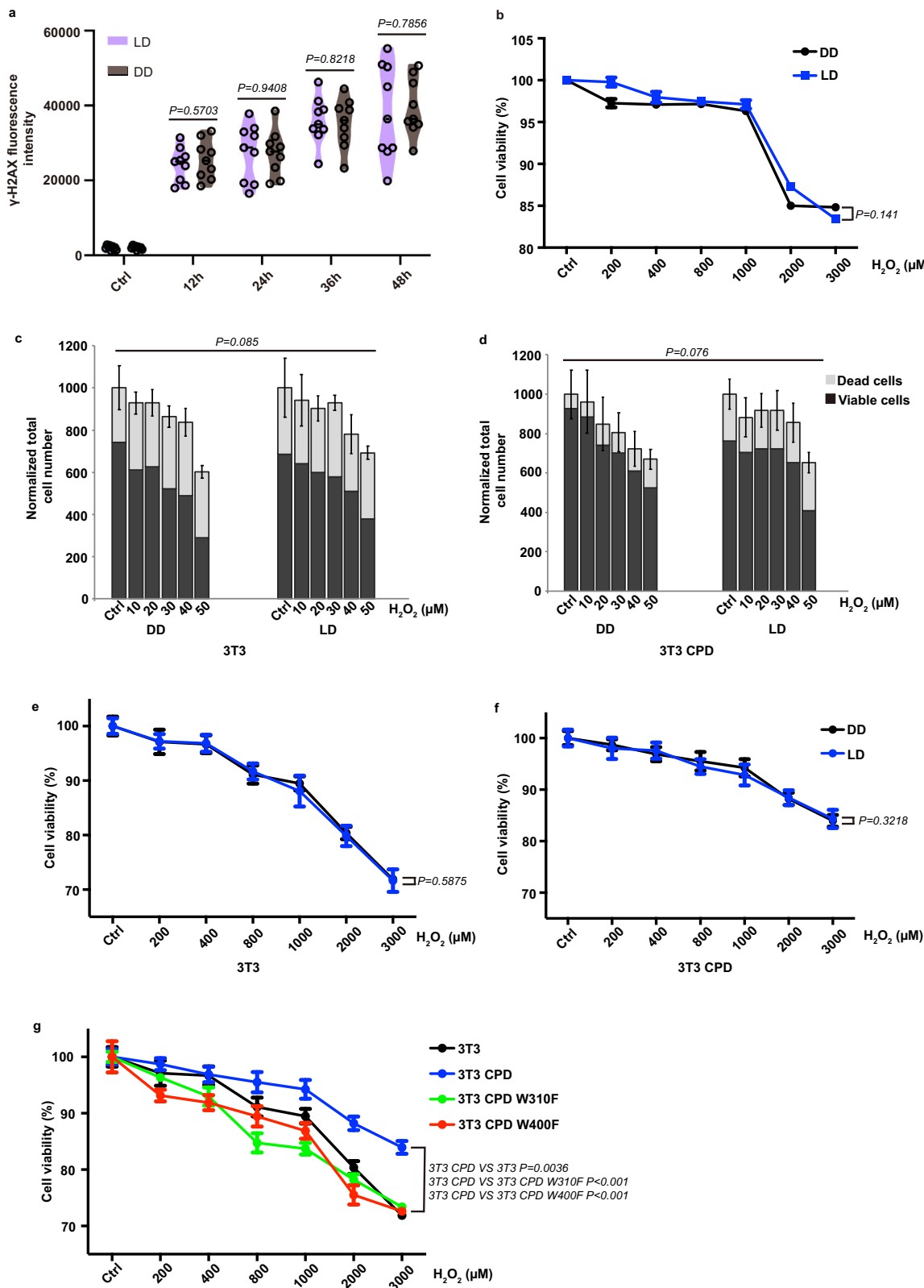

have been documented for CPDphr binding to CPD. Does the light-independent repair of CPD and 8-OHdG damage by CPDphr represent an enzyme-catalysed reaction akin to photoreactivation itself? It would seem unlikely given the structural differences between these two types of DNA damage. Another possibility is that in darkness, CPDphr might bind to this DNA damage via a distinct region of the tertiary protein structure and thereby serve as a recognition complex which would

then recruit other components of the DNA repair machinery belonging to repair pathways such as NER or BER. Clearly a priority for future studies will be the identification and characterization of the DNA repair systems that cooperate with CPDphr to enable this light-independent repair function of ROS-induced DNA damage.

Based on our results, we propose the following model for DNA damage repair involving two distinct functions of CPDphr (Fig. 6).

**Fig. 5 | Impact of light on ROS-induced DNA damage repair. a** Violin plot quantification of immunofluorescence assays of γ-H2AX levels in medaka WT cells transiently treated with 400 μM $H_2O_2$ and subsequently left to recover under light-dark cycle (LD) or constant darkness (DD) conditions. Cells were fixed at various timepoints indicated on the x-axis. Mean values of γ-H2AX fluorescence intensity are indicated by black horizontal lines and individual data points ($n = 9$ biologically independent samples) are shown as black hollow circles. **b** Cell viability assay of medaka WT cells exposed transiently to $H_2O_2$ concentrations from 200 μM to 3000 μM and subsequently left to recover under LD or DD conditions. Mean percentage ± SEM ($n = 8$ biologically independent samples) of cell viability with respect to untreated cells are indicated on the y-axis, while $H_2O_2$ concentrations are denoted on the x-axis. **c, d** Results of AHM assays of 3T3 and 3T3 CPD cells exposed transiently to a range of $H_2O_2$ concentrations from 10 μM to 50 μM, and then allowed to recover under LD or DD conditions. Total numbers of dead and viable cells are plotted as means ± SEM ($n = 4$ biologically independent samples) normalized in relation to untreated cells (Ctrl) on the y-axis, while $H_2O_2$ concentrations

are indicated on the x-axis. **e, f** Cell viability assay of 3T3 and 3T3 CPD cells transiently exposed to a range of $H_2O_2$ concentrations from 200 to 3000 μM followed by recovery under LD or DD conditions. Mean percentage ± SEM ($n = 8$ biologically independent samples) of cell viability with respect to untreated cells is plotted on the y-axis, with $H_2O_2$ concentrations on the x-axis. **g** Cell viability assay of various 3T3 cell clones (3T3, 3T3 CPD, 3T3 CPD W310F, 3T3 CPD W400F) transiently exposed to a range of $H_2O_2$ concentrations, from 200 to 3000 μM and subsequently left to recover under DD conditions. Mean percentage ± SEM ($n = 8$ biologically independent samples) of cell viability with respect to untreated cells is plotted on the y-axis, while $H_2O_2$ concentrations are indicated on the x-axis. All assays were performed at least 3 times, independently. For the cell viability assays (**b**–**g**), representative data is shown. The statistical test used for (**a**) is student's t-test, while the analysis for (**b**–**g**) is two-way ANOVA analysis. Statistical differences (P values) are annotated in each panel. Source data are provided as a Source Data file.

Under sunlight exposure, CPD is generated via two mechanisms; directly via the effects of UV radiation as well as indirectly via increasing intracellular levels of ROS which also triggers 8-OHdG formation. Visible wavelengths of sunlight serve as a source of energy for CPDphr to repair CPD damage by photoreactivation. However, CPDphr also binds to CPD and 8-OHdG damage and via a light-independent mechanism, serves to target other repair mechanisms to the damaged sites. Alternative models implicating an indirect function of CPDphr in modulating redox homeostasis could theoretically be envisaged. However, given the ability of CPDphr to interact directly with CPD and 8-OHdG damage, and its well-documented function in DNA repair, the more parsimonious explanation would seem to be a direct interaction with other DNA repair mechanisms. In a constant dark environment, CPD and 8-OHdG damage is generated only by endogenous ROS levels and in the absence of photoreactivation only the light-independent CPDphr repair function still operates.

The cavefish *P. andruzzii* is remarkable compared with other cavefish species in that it exhibits the most extreme troglomorphic phenotypes such as complete eye loss and absence of body pigmentation. This reflects the evolution of this species that is predicted to have occurred over the course of 3 million years completely isolated from surface bodies of water beneath the Somalian desert[37]. The other widely studied cavefish species *Astyanax mexicanus* shows less pronounced troglomorphic phenotypes due to a shorter period of isolation in the cave environment (20.000 to 200.000 years[38]) as well as some contact with normal epigean forms of this species, which live in interconnected surface water. Interestingly, *A. mexicanus* has significantly elevated basal levels of CPDphr expression and exhibits much lower levels of DNA damage compared with surface forms[39]. Although initially surprising given the absence of light to drive photoreactivation, this finding would be entirely consistent with CPDphr possessing a light-independent repair function. The cavefish *P. andruzzii* shares the loss of the majority of its photolyase photoreactivation function with placental mammals. One popular theory to explain several adaptations of the visual systems of placental mammals that are characteristic of nocturnal animals is the so-called "nocturnal bottleneck theory". This theory predicts that to escape the predatory dinosaurs, the dominant taxon during that era, the predecessors of placental mammals adopted a predominantly nocturnal and subterranean lifestyle[40]. These changes resulted in reduced reliance on solar exposure and eventual loss of photolyase genes. In the case of placental mammals, even the CPDphr gene was lost suggesting a plausible relaxation scenario for photolyases in general.

Groundwater colonization, followed by surface desertification and the complete isolation of *P. andruzzii* in a subterranean dark environment, likely led to a reduction in effective population size, potentially reflected in a low level of genomic variation. In this scenario, the possibility that the conservation of the CPDphr gene

sequence is due to chance, either through genetic drift following groundwater colonization or the sampling of individuals with low genetic variation in this specific genomic region, cannot be ruled out. However, the complete lack of variation across the 14 kb CPDphr gene observed in 14 *P. andruzzii* chromosomes appears extreme, especially when contrasted with the eight polymorphic positions identified within the 5 kb 6-4phr gene across 12 chromosomes. This is further underscored by the total or partial loss of function observed in several genes involved in light-related signalling pathways[37,41,42]. An alternative explanation involving purifying selection would require identifying a putative selective agent: our results indicate that in particular subterranean environments, an elevated level of ROS may be a good candidate, imposing constraints favouring CPDphr function. This work illustrates how the evolution of the DNA repair repertoire under extreme environmental conditions can provide new insight into these mechanisms and the role they play in whole-organismal biology.

## Methods

### Fish maintenance and dissection of tissues

Wild type (iCab) and photolyase mutant medaka (*Oryzias latipes*) were maintained in the fish facility of the Institute of Biological and Chemical Systems, Biological Information Processing (IBCS-BIP) at the Karlsruhe Institute for Technology (KIT). The mutant photolyase medaka lines were generated using CRISPR-Cas9-based mutagenesis at Osaka University, Japan as previously described[12]. The fish were raised in a water circulation system at a temperature of 26 ± 1 °C, with a light cycle of 14 h of light and 10 h of darkness. The experiments performed with fish were conducted in compliance with the European Legislation for the Protection of Animals used for Scientific Purposes (Directive 2010/63/EU) (General license for fish maintenance and breeding: Az.: 35-9185.64/BH KIT IBCS-BIP, Karlsruhe Institute of Technology (KIT)) and adhered to the animal protection standards of Germany. The permission to perform experiments with these medaka mutant lines was approved under the license: 35-9185.81/G-132/21 and 35-9185.81/G-149/23 KIT IBCS-BIP.

In the medaka fin clip experiments, the fish were anesthetized using 0.02% Tricaine (Sigma Aldrich) diluted in embryo rearing medium (ERM) for 2–3 minutes. Subsequently, a small portion of the fin was cut using a scalpel. To facilitate recovery, the fish were transferred to ERM supplemented with methylene blue. The excised fin segments were subjected to three successive washes in 1× PBS and Leibovitz's L-15 medium, and thereafter treated according to the experimental design. To dissect the brain and liver tissues, medaka fish were euthanized by immersing them in ice water. The tissues were then dissected and rapidly frozen in liquid nitrogen. Subsequently, they were stored at -80 °C until further processing.

Adult cavefish *Phreatichthys andruzzii* were obtained from a colony maintained at the University of Ferrara (General license for fish

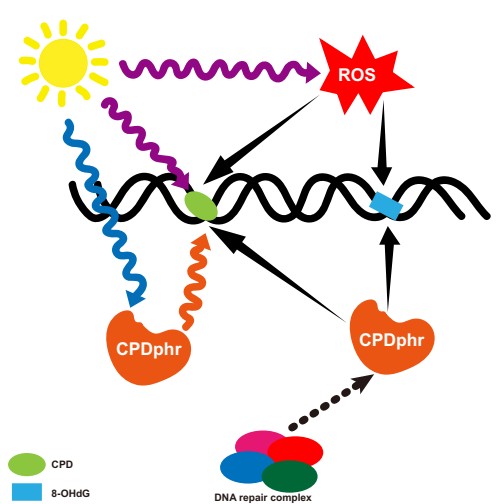
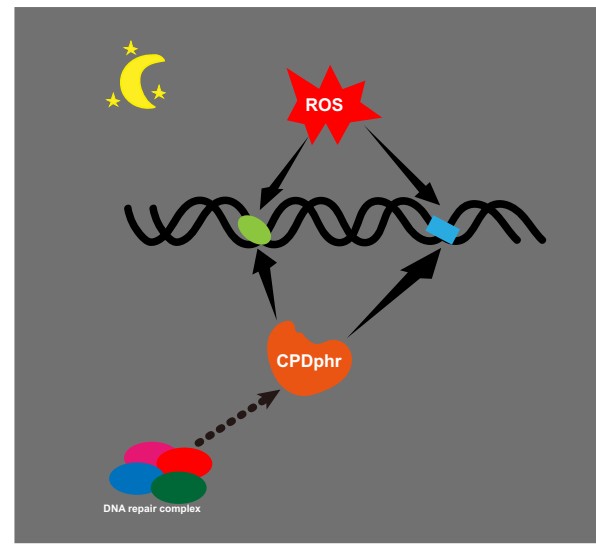

**Fig. 6 | Light-dependent and -independent functions for CPDphr.** Schematic representation of our model for CPDphr function under sunlight exposure (left panel) and in darkness (right panel). Under sunlight, UV exposure induces covalent modifications of adjacent pyrimidine bases and thereby generates CPD photo-products. In parallel, UV exposure also leads to increases in intracellular levels of ROS which indirectly generates CPD as well as other types of DNA damage including 8-OHdG. Visible wavelengths of light serve as a source of energy for CPDphr to repair CPD damage by photoreactivation. In a light-independent mechanism, which, like photoreactivation, relies on the integrity of the three-tryptophan Electron Transfer Chain at the active site of this protein, CPDphr also binds to CPD and 8-OHdG damage, and thereby we speculate may serve as a tag for other repair mechanisms to operate. Under the constant dark subterranean environment, CPD and 8-OHdG damage is generated only by endogenous ROS levels, and while CPDphr photoreactivation function is absent, its light-independent repair function still operates.

maintenance and breeding: no. 18/2017-UT). Cavefish were kept in darkness except during food administration and aquaria maintenance. All aquaria were equipped with mechanical and biological filters. The temperature of the facility was maintained at $28 \pm 1\,°C$ by means of an automatic air conditioning system. To collect fin clips for DNA extraction, cavefish were anaesthetized in 0.02% Tricaine (Sigma Aldrich) solution. After excision, each fin clip was recovered with a sterile pipette tip and transferred into a tube filled with absolute ethanol. The permission to perform experiments with cavefish was approved by the Italian Ministry of Health (aut. N. 890/2016-PR) and were conducted in compliance with European (Directive 2010/63/EU) and Italian legislation.

The sex or gender of vertebrate animals and cell lines was not considered in our experimental design.

### Fish and mammalian cell culture and transfection
The medaka wild type and mutant cell lines derived from medaka embryos (CPD/DASH/6-4 WT/KO) were cultured in Leibovitz's L-15 medium (Gibco) supplemented with 20% Fetal Bovine Serum (FBS) (Gibco), 100 units/ml penicillin, 100 μg/ml streptomycin and 100 μg/ml gentamicin (Gibco). All these cell lines were incubated in an atmospheric $CO_2$, non-humidified cell culture incubator at 26 °C.

The mouse cell line NIH 3T3 and the stably transfected cell lines that were derived from it were cultured in DMEM medium (Gibco) supplemented with 10% Fetal Bovine Serum (FBS) (Gibco), 100 U/ml penicillin, 100 μg/ml streptomycin (NIH 3T3 cells) or 250 μg/ml neo-mycin (stably transfected 3T3 cell lines) (Gibco). These cells were maintained in a 5% $CO_2$, humidified cell culture incubator at 37 °C.

3T3 cells were transfected using FuGene HD transfection reagent (Promega) according to the manufacturer's instructions. The ratio of FuGene (μl) to plasmid DNA (μg) was 4:1, and cells were incubated with the FuGene / DNA mixture overnight at 37 °C.

### Light sources and H₂O₂ treatment
All experiments involving light exposure were conducted using one of the following light sources at a constant temperature of 26 °C (fish cells) or 37 °C (mouse cells): laboratory UV-C light (VETTER GmbH, 254 nm), white light emitting diodes (LED, Kopa, 440–690 nm) and monochromatic red light emitting diodes (LED, Kopa, 665 nm). Due to the weak penetration of UV-C through aqueous solutions, the medium was removed from the plate before UV-C treatment and replaced immediately afterwards. Hydrogen peroxide ($H_2O_2$, Sigma Aldrich) treatment was carried out according to the manufacturer's instructions, by simply adding $H_2O_2$ to the medium in the wells and incubating for various time periods depending on the experimental design. Subsequently, the $H_2O_2$-containing medium was discarded and replaced by fresh medium for recovery.

### Immunofluorescence analysis
Immunofluorescence assay for γ-H2AX levels in cells and fin clips was performed as previously described[43] with modifications. Cell cultures were seeded in a 24-well plate and after 1 day were treated with $H_2O_2$ for 1 h in constant darkness followed by a recovery period of 1 h also in constant darkness. Fin clips were treated with $H_2O_2$ for 1 h in constant darkness and then immediately harvested for the immuno-fluorescence assay. The primary and secondary antibodies employed are listed in Supplementary Table 3. Samples were imaged using a confocal microscope (Leica TCS SP5). The acquired images were viewed and the degree of DNA damage caused by oxidative stress as measured by nuclear γ-H2AX staining was quantified by Fiji/ImageJ software.

### Cell viability assay
$8 \times 10^3$ cells (medaka cells) or $1.5 \times 10^3$ cells (mammalian cells) were plated per well of a 96-well plate for automated high-throughput microscopy (AHM) measurement of cell number and the proportion of living and dead cells. The cells were incubated in darkness for 2 days and then treated with various doses of $H_2O_2$ for 1 h in darkness. Subsequently, the cells were allowed to recover for 1 day in darkness and then the microscopy analysis was performed according to a previously described protocol[44,45]. Following staining with Hoechst 33342 and propidium iodide (PI) at 0.3 g/ml and 0.5 g/ml, respectively, bright

field (BF) and fluorescence images were obtained from four areas of each well by an automated Olympus IX81 fluorescence microscope. The acquired images were processed by using scan^R analysis software (version 2.7.3, Olympus, Hamburg, Germany) to identify the number of living and dead cells.

An MTT (3(4,5dimethylthiazol-2yl)2,5diphenyltetrazolium-bromid, Sigma Aldrich) assay was employed to determine cell viability as previously described[46]. Specifically, cells were seeded at a density of $3 \times 10^4$ cells/well (fish cells) or $1 \times 10^4$ cells/well (mammalian cells) in a 96-well plate and maintained in constant darkness for 2 days. Afterwards, cells were exposed to various concentrations of $H_2O_2$ for 1 h in darkness or exposed briefly to UV-C light. Following $H_2O_2$ treatment cells were allowed to recover for 1 day in darkness while UV-exposed cells were allowed to recover for 2 days in darkness or during exposure to a 12 h light: 12 h dark cycle. Subsequently, 0.5 mg/ml MTT dissolved in L-15 medium was added for 4 h at 26 °C (fish cells) or 37 °C (mammalian cells). DMSO was subsequently added to dissolve the purple formazan crystals and the plate was measured by spectrometry using a test wavelength of 590 nm with a reference wavelength of 620 nm on a SpectraMax iD3 Microplate Reader.

### Cloning, mutagenesis and establishment of gain-of-function 3T3 cell line

For eukaryotic expression, the coding sequence of zebrafish CPDphr was cloned into the pCS2-MTK expression vector[9]. The CPDphr W310F and W400F mutants were generated by using the Q5 Site-Directed Mutagenesis Kit (New England Biolabs) according to the manufacturer's instructions. The primers used are shown on Supplementary Table 4.

For transfection, 3T3 cells were seeded at a density of $4 \times 10^5$ cells/well in a 6-well plate and thereafter were co-transfected with the CPDphr expression vectors and an empty pcDNA vector carrying a neomycin resistance cassette using the Fugene transfection system. Following initial neomycin selection, viable cells were trypsinized and reseeded into individual wells of a 96-well plate, to ensure that each well of the 96-well plate carried only 1 cell. Eventually, after several weeks, multiple single clone cell lines expressing various amounts of photolyase proteins were obtained.

To verify ectopic CPDphr expression in each clone, a western blotting analysis was performed by following the manufacturer's instructions. After primary and secondary antibody incubation, the Clarity Western ECL substrate (Bio-Rad) was applied to the PVDF membrane and the membrane was visualized using the ChemiDoc™ Imaging System (Bio-Rad). The antibodies used are listed in Supplementary Table 3. Images were viewed and evaluated using ImageLab™ software (Bio-Rad).

### ELISA assay

ELISA (Enzyme-Linked ImmunoSorbent Assay) assays were employed to directly measure the levels of DNA damage following UV-C and oxidative stress exposure. An OxiSelect™ UV-Induced DNA Damage ELISA Kit and OxiSelect™ Oxidative DNA Damage ELISA Kit (Cell Biolabs) were used to quantify CPD and 8-OHdG levels, respectively. Cells were plated at a density of $6 \times 10^5$ cells/well (CPD quantification) in a 6-well plate or $2 \times 10^6$ cells/dish (8-OHdG quantification) in a 10 cm petri dish and cultured in darkness for 2 days. Brain, liver and fin clip tissues were dissected from the iCab and CPDphr mutant fish. Samples were then treated according to the experimental design and the kit manufacturer's instructions. Following treatment, cells were harvested and genomic DNA from cells and tissues was extracted using the GeneJET Genomic DNA Purification Kit (Thermo Fisher). In the case of 8-OHdG quantification, samples were completely digested to generate free nucleotides by nuclease P1 treatment and dephosphorylated by alkaline phosphatase. Subsequently, samples were applied to the plate supplied with the kits.

### Purification and EMSA binding assay of photolyase protein

CPDphr was expressed in the *E. coli* Er2566 bacterial system and the bacteria were cultured in 2 L LB medium with Ampicillin at 37 °C until the culture reached an $OD_{600}$ of 0.6 to 0.8. Then, the expression was induced by the addition of 100 μM isopropyl β-*D*-thiogalactopyranoside (IPTG) and incubated overnight at 18 °C. After centrifugation and resuspension in basic buffer (50 mM Tris, 5 mM EDTA, 300 mM NaCl, 10% glycerol, pH 7.8), the cells were disrupted using a French press and the protein was precipitated by addition of ammonium sulphate. Subsequently, the protein sample was applied to a wash buffer equilibrated column packed with $Ni^{2+}$-NTA agarose matrix (Qiagen) and eluted with elution buffer (50 mM Tris, 250 mM imidazole, 300 mM NaCl, 10% w/v glycerol, pH 7.8). The eluate was concentrated by ammonium sulphate precipitation and the protein was resuspended in basic buffer[47,48]. The identification and purity of the eluted proteins were tested by electrophoresis of the purified proteins through a SDS-PAGE (Sodium dodecyl-sulfate polyacrylamide gel electrophoresis) gel as previously described[48].

For the Electrophoretic Mobility Shift Assay (EMSA), a probe with the following sequence: 5′-(AGCTACCATGCCTGCACGAA<u>TT</u>AAGC AATTCGTAATCATGG TCATAGCT)-3′ and its complementary strand: 5′-(AGCTATGACCATGATT ACGAATTGCTTAATTCGTGCAGGCATGG-TAGCT)-3′ were synthesized (Sigma Life Science) and both strands were labelled with biotin at their 3′-termini using the Biotin 3′ End DNA Labelling Kit (Thermo Scientific). The end-labelled probes were annealed by heating at 90 °C for 5 min followed by slow cooling to 25 °C over 1–2 h. This labelled double-strand probe was exposed to UV-C light for 6 h in order to generate CPD photoproducts from the two adjacent thymidines located in the middle of the sequence (underlined TT bases in the sequence above). The 8-OHdG modified probe with an identical sequence and biotin labelling at the 3′ ends, but incorporating a single modified guanine (underlined G in the sequence above), was synthesized by Metabion and annealed again following the manufacturer's instructions.

The EMSA assay was performed by using the LightShift® Chemiluminescent EMSA Kit (Thermo Scientific) with biotin-end-labelled oligonucleotide probes and bacterially produced CPDphr protein. The resulting acrylamide gels were electrophoretically transferred to nylon membranes (Carl Roth) and the transferred DNA was crosslinked to the membrane by exposure to UV-light ($120 \ mJ/cm^2$) according to the manufacturer's instructions. Thereafter, biotin-labelled DNA was detected by Chemiluminescence using the ChemiDoc™ Imaging System (Bio-Rad).

### Sequence analysis

To verify whether, at the genomic level, the sequences of CPDphr and 6-4phr were conserved and enriched in null-function mutations, respectively, as suggested by previous transcript analyses[9], two assays were designed. Specifically, amplicons spanning the entire coding sequences were obtained by long PCR starting from about 50 ng of genomic DNA extracted from fin clips by means of the DNeasy blood and tissue Kit (Qiagen). A 14 Kb amplicon spanning the CPDphr locus of six *P. andruzzii* individuals was obtained using the primer pair listed in Supplementary Table 2, and the following PCR conditions: 1 U of GSTAR enzyme (Takara) in a 50 μl reaction; 29 cycles (98 °C/10″ + 56 °C/15″ + 68 °C/10′). Short-read genomic libraries were constructed using a Nextera DNA Flex Library Prep Kit (Illumina) according to the manufacturer's protocol. Libraries were sequenced paired-end (2 × 150 bp) on an Illumina NovaSeq 6000 System using a 300-cycle NovaSeq 6000 S1 Reagent Kit v1.0. The reference sequence was reconstructed by means of a de novo assembly starting from the reads of one individual using SPAdes[49]. The remaining individual sequences were aligned to the newly assembled reference sequence using the Geneious aligner (Geneious v8.1.8[50]) with intermediate parameters

between high and medium sensitivity: five iterations; maximum mismatches per read 40%; maximum gap per read 20%; maximum gap size 50 bp. A 5 Kb amplicon spanning the 6-4phr locus of six individuals was obtained with the primer pair listed in Supplementary Table 2, following the same PCR reaction described above, except with a 6' extension. Since direct haplotype phasing is difficult with short reads, this polymorphic locus was sequenced with nanopore technology (Oxford Nanopore Technologies, ONT). Following AMPure beads (Beckman Coulter) purification (beads concentration: 1.8x; 10' shaking; two cleaning steps with 70% ethanol), the 6-4phr amplicons were then sequenced with minION ONT R9 flow cells after library preparation using a standard procedure for the Ligation Sequencing Kit (SQK-LSK109). Pod5 reads were basecalled using Guppy[51], and the resulting fastq files were *de-novo* aligned with Geneious aligner (Geneious v8.1.8[50]). Variant calling was performed with Geneious and each polymorphic position was visually inspected. Ambiguous sites were further confirmed by Sanger sequencing.

### Quantification and statistical analysis

All the data were calculated and represented as means ± SEM. Additionally, all data were plotted by using GraphPad Prism 10 (GraphPad Software Inc.) and Excel 2019 and analyzed by SPSS Statistics 19.0 (IBM). To identify statistically significant differences, either a Student's *t*-test or an analysis of variance (ANOVA) was employed, followed by Sidak's multiple comparison post-test. Statistics were considered significant when the *p* value was less than 0.05.

### Reporting summary

Further information on research design is available in the Nature Portfolio Reporting Summary linked to this article.

## Data availability

All data acquired or analyzed for this work are included in this published article or provided as source data files. Source data are provided with this paper.

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

## Acknowledgements

This work was mainly supported by funding from the NACIP programme of the Helmholtz Association as well as by the DFG project FO 549/3-1. N.S.F. and D.V. were funded by the NACIP programme of the Helmholtz Association and H.L. was funded by the DFG project FO 549/3-1 and Shandong Mingru Biotech Co., Ltd. C.B. and S.F. were supported by research grants from the University of Ferrara (FAR2023 and FAR2024). We acknowledge support by the Deutsche Forschungsgemeinschaft and the open access publishing fund of Karlsruhe Institute of Technology. We are grateful for the support of the International Zebrafish and Medaka FELASA accredited Course (IZMC_F059/17) and we thank Ettore Fedele for help in gene sequencing analyses. Figure 1 was partially created using BioRender.com. We thank Haiyu Zhao, Olivier Kassel, Yuhang Hong, Alessandra Boiti, Felix Loosli and Rima Siauciunaite for support and discussion and Nathalie Geyer, Nadeshda Wolf, Christina Münzing, Nina Spohrer and Gero Kaeser for excellent technical assistance.

## Author contributions

Conceptualization and design of experiments: H.L., S.F., D.V., C.B., N.S.F. Performing experiments: H.L., C.S., G.D.M., S.F., S.F.D. Writing and revision of the manuscript: H.L., G.D.M., S.F., T.T., C.W., D.V., T.L., C.B., N.S.F.

## Funding

## Competing interests

The authors declare no competing interests.
