## [Transparent Peer Review file · Nature Communications]

Conservation of dark CPD photolyase function in blind cavefish

Corresponding Author: Professor Nicholas Foulkes

Version 0:

Reviewer comments:

Reviewer #1

(Remarks to the Author)

The manuscript: Conservation of “dark” CPD photolyase function in blind cavefish“ by Li et al present an exciting and well-executed study that leverages the blind cavefish *Phreatichthys andruzzii* to uncover a previously unrecognized function of CPD photolyase (CPDphr) beyond its canonical role in photoreactivation. While *P. andruzzii* has lost other photolyases due to relaxed selection in perpetual darkness, CPDphr remains conserved, prompting the authors to explore whether it serves a function independent of light exposure. Their findings reveal a novel role for CPDphr in oxidative DNA damage repair, particularly in countering 8-hydroxydeoxyguanosine (8-OHdG) lesions, a discovery with broad implications for DNA repair biology.

This study exemplifies the power of cavefish as a model for uncovering counterintuitive biological insights that lead to new functional hypotheses. By investigating a paradoxical retention of a photolyase gene in a light-deprived organism, the authors provide strong evidence that CPDphr plays a crucial role in mitigating oxidative stress in extreme environments. This not only enhances our understanding of subterranean adaptations but also challenges the traditional view of photolyases as solely light-dependent DNA repair enzymes. The multi-system approach using different cell lines and in vitro and in vivo approaches over a broad range of species is innovative and convincing, the methodology sound.

Overall, this study is an excellent demonstration of how extreme environmental models can reveal fundamental biological principles. By capitalizing on a unique evolutionary scenario, the authors not only advance our understanding of cavefish adaptations but also contribute to the broader field of DNA repair biology. I really enjoyed reading it.

Reviewer #2

(Remarks to the Author)

Li et al. investigates the evolutionary conservation and functional significance of CPD photolyase (CPDphr), a DNA repair enzyme, in the blind cavefish (*Phreatichthys andruzzii*). The authors convincingly demonstrate that CPDphr possesses a previously unrecognized ability to repair DNA damage in a light-independent manner, specifically targeting CPD and 8-OHdG. The integration of multiple approaches including CRISPR-generated loss-of-function in vitro and in vivo mutants, ectopic expression assays in mammals, cell survival assays, and purification of CPDphr to test binding affinity is impressive. This study represents a significant, carefully conducted contribution to the literature that provides novel insights into the evolution and adaptation of DNA repair mechanisms in organisms, especially those living in perpetual darkness.

Substantial critiques

1) The assertion that the ‘dark’ function of CPDphr is adaptive in cave environments (lines 102-103 and 405-408) and that selection to maintain a functional copy resulted in the high sequence conservation compared to other UV light damage response genes would benefit from additional evidence.

Please provide information on the colony (how many generations in captivity, how many breeding individuals). This is important for putting the genetic variation into context. Could the conservation/low sequence diversity be related to chance of collecting and then breeding individuals that randomly had little genetic diversity in that genomic region in the first place? Is the difference in sequence diversity between CPDphr and 4-6PPs reflected in wild samples? When only comparing two genes, it’s difficult to tell which gene may be driving the difference (e.g. is LOF of 6-4 PP, high conservation of CPDphr, or

both driving the differences). Additional explanations for the lack of diversity of CPDpnr are not explored and limited population genetic data from the lab colony are not sufficient to say that selection is the cause of the observed lack of sequence diversity. For example, tight linkage to another gene under selection could also result in the observed lack of diversity in CPDpnr or perhaps most of the genome has very low diversity and the LOF alleles and relaxed purifying selection result in high sequence diversity. Context about whether the genes are of similar length or comparison to other genes could be a simple way to boost support for this observation.

2) The use of barplots is not ideal as they obscure the distribution of the actual data.

<https://www.nature.com/articles/nmeth.2807> Please use data visualization (e.g. box plots, violin plots) that can better represent the variation.

3) This work applies many techniques, and a broad audience may find it hard to keep track of the impressive set of assays use help test the central hypothesis that CPDpnr functions in the dark and has a large role repairing DNA damage beyond UV damage.

One option to add clarity could be that when a specific cell line vs whole tissue is introduced, rationale for why using that cell line or tissue is included (e.g. lines 85-86,145-148, 216-218). Authors do a nice job introducing mammalian cell line, but the medaka vs zebrafish, cell vs fin clip could also use similar treatment. Such additions would help readers follow along with the analyses and provide justification for using such cell lines from organisms that aren't the focal cave fish *P. andruzzii*, while highlighting the extensive capacity for 'dark' CPDpnr activity across diverse organisms/cellular contexts.

Minor comments

1. L60. Please avoid the term "higher" vertebrates, it implies a hierarchy in evolution that doesn't scientifically exist.
2. Lines 178-206 CPDpnr transfected 3T3 vs non-transfected 3T3. I am not experienced with many of the assays done in this study, but is the most appropriate control in this comparison supposed to be between two transfected cell lines (one with the gene of interest, one with a sort of control gene not expected to have a similar function) to rule out a role of the transfection process in the cells ability to handle stressful conditions that follow? If that's not a standard control in these types of assays then perhaps one additional sentence about how the transfection process isn't expected to skew the results would help a broad audience better understand the assay (or if it is a more ideal control but practicality/resources didn't allow, then a caveat sentence would be warranted).
3. Lines 216-223: In the comparison of iCab medaka to CPDpnr medaka(?) fish – is the only big difference expected between these two lines the knockdown mutation of CPDpnr? Some explanation about the differences between these two lines should be mentioned to help readers interpret the resulting differences in CPD levels being solely due to having a functional CPDpnr or not.
4. Figure 2e: From eyeballing the counts across the two panels, it appears that far fewer cells (dead or alive) were accounted for in right panel compared to the left panel despite even numbers of cells going into the cell viability assay being mentioned in the methods. Is this an additional sign that the cells without functional CPDpnr were so damaged/dead from the H202 exposure that they couldn't be accounted for during the counting process? Or something else? Could be helpful to explain this pattern when talking about lower alive cell counts in the results section.
5. Supplementary Figures 3, 5, 6, 8, 7, 9. Can you please add a biological interpretation to the legend for the reader (i.e. what is the take home message from this?)
6. Figure S4. How do you interpret the more cell viability in LD than DD? Can you explain a bit more for the reader?
7. Figure 2. Please specify whether these were LD or DD conditions.
8. Figure S8. Is there a control for LD? It doesn't appear that the immunofluorescence in the panel (b) for 36hrs and 48hrs matches what is shown in the bargraph for (a). Perhaps more explanation is needed?

Reviewer #3

(Remarks to the Author)

This study, titled "Conservation of 'Dark' CPD Photolyase Function in Blind Cavefish" by Li et al. explores how DNA repair mechanisms evolve in response to environmental pressures, particularly in organisms that have adapted to life in complete darkness. DNA damage caused by ultraviolet (UV) radiation is typically repaired by photoreactivation, a process that relies on photolyases—highly conserved enzymes that use light to reverse UV-induced lesions, such as cyclobutane pyrimidine dimers (CPDs) and 6-4 photoproducts (6-4PPs). However, placental mammals and certain cave-dwelling species, including the blind cavefish *Phreatichthys andruzzii*, have lost functional photolyase genes. Previous research from the Foulkes and Bertolucci laboratories has demonstrated that *P. andruzzii* has undergone significant modifications to its DNA repair systems after millions of years in complete darkness. Specifically, these studies revealed that the *P. andruzzii* cavefish has lost photoreactivation capability due to mutations in 6-4pnr and DASHpnr photolyase genes, as well as disruptions in the D-box enhancer element, which regulates DNA repair responses to light and oxidative stress. Despite these losses, the CPD photolyase (CPDpnr) gene remains highly conserved, raising the question of whether it has been repurposed for a novel function.

In this study, the group builds off past work to address what non-light-mediated function CPDpnr may have in fish. Using CRISPR-generated medaka models with loss of CPDpnr function and 3T3 mammalian cell lines transfected with zebrafish CPDpnr, they show that loss of CPDpnr results in elevated dsDNA damage, whereas the addition of zCPDpnr buffers fibroblast lines from oxidative damage, as measured by γ -H2AX staining. These data suggest that CPDpnr, but not other photolyases, may ensure cell survival by protecting against DNA damage. Further, they show that CPDpnr protects against

CPD accumulation in the liver but not in the brain, while also protecting against 8-OHdG accumulation in both brain and liver. EMSA binding assays further reveal that CPDphr directly binds to 8-OHdG-modified DNA, supporting its role in oxidative stress-related repair. Lastly, the group demonstrates that these protective functions are not dependent on light-dark cycles but still require the “tryptophan residue triad” important for CPDphr electron transfer. This suggests that while CPDphr has evolved to function in darkness, it still relies on a conserved electron transfer mechanism. The group presents a model proposing that CPDphr acts on both light-induced and ROS-induced DNA damage in other species, whereas in *P. andruzzii*, its function may be exclusively ROS-related due to the loss of photoreactivation capacity.

The manuscript is well written, the statistics are appropriate, and the findings are important as they challenge the long-standing view that photolyases are exclusively light-dependent enzymes. By showing that CPDphr plays a critical role in repairing oxidative DNA damage even in total darkness, the study expands our understanding of DNA repair mechanisms and provides an example of how evolutionary pressures can drive the functional repurposing of conserved genes. This work has broad implications for evolutionary biology, DNA repair research, and even biomedical applications related to oxidative stress-induced damage.

I have only a few concerns/questions:

- Tissue-specific discrepancy: The study reports that CPDphr prevents CPD accumulation in the liver but not the brain while protecting against 8-OHdG in both. The reason for this discrepancy is presented but not fully explored. Does the brain have alternative repair mechanisms? Is the liver more susceptible to ROS-induced CPD formation? Addressing this gap would provide a clearer picture of how CPDphr is selectively active across tissues.
- 3T3 cells are useful tools for studying gain of function, though it's not clear why this line was chosen. Might a better line be one that completely lacks all UV-damage repair pathways (XP-A fibroblasts). This could be used to determine whether the observed protection is independent of existing repair mechanisms.
- The study attributes the protective effect of CPDphr primarily to its direct role in DNA repair, specifically in repairing CPD and 8-OHdG lesions under oxidative stress. However, there are several alternative mechanisms that could contribute to the observed protective effects. Specifically, CPD-phr could interact with oxidative stress response pathways or be broadly involved in modulating redox homeostasis. It's unclear to me how a direct link to DNA damage repair complex is deduced as presented in Fig 6.

Minor points

The manuscript is very well written, but a few grammatical errors can be cleaned up:

Line 368–370: “Based on our results, we propose the following model for DNA damage repair involving dual functionality of CPDphr (Fig. 6).” It took me a while to understand what this sentence said. I'd suggest something simpler like “involving two distinct functions of CPDphr”

Line 568–570: “Fin clips were recovered with a sterile pipette tip and transfer into a tube filled with absolute ethanol.” Should be “and transferred.”

Line 570–572: “The permission to perform experiments with cavefish was approved by Italian Ministry of Health (aut. N. 890/2016-PR).” Should be “by the Italian Ministry of Health”

Reviewer #4

(Remarks to the Author)

Version 1:

Reviewer comments:

Reviewer #1

(Remarks to the Author)

I have no further comments

Reviewer #2

(Remarks to the Author)

All of my comments were addressed. This is really nice contribution.

Reviewer #3

(Remarks to the Author)

The authors have fully addressed my concerns raised in the initial review. Specifically, they have provided a thoughtful

explanation for the observed tissue-specific discrepancies in DNA damage (notably, CPD accumulation in liver but not brain) in the discussion, citing established differences in DNA repair mechanisms and oxidative stress responses across tissues. They have also clearly justified their use of 3T3 cells for gain-of-function studies, explaining their suitability due to the lack of endogenous photolyase activity, while acknowledging that future work in alternative cell models (e.g., XP-A fibroblasts) would be valuable. Lastly, the authors did a good job clarifying how CPD photolyase might function in the dark. They showed direct interaction with damaged DNA—specifically 8-OHdG lesions—using electrophoretic mobility shift assays. While their main focus remained on DNA repair, they also acknowledged the possibility of broader roles, like involvement in redox regulation, and updated both the discussion and summary figure to reflect that.

In summary, the revised manuscript addresses my concerns, and I support the publication of this work in its current form.

Rebuttal Letter

Reviewer 1

We would like to thank this Reviewer for their extremely positive assessment of our work and for sharing our enthusiasm about the power of comparative studies involving extreme environmental models. Thank you!

Reviewer 2

We would like to thank this reviewer for their very positive verdict on our work. They noted that our „*integration of multiple approaches.....is impressive*“. Also, they felt that our study represents „*a significant, carefully conducted contribution to the literature that provides novel insights into the evolution and adaptation of DNA repair mechanisms.....*“

This reviewer raised a series of constructive criticisms that we fully agree with - and which we have now addressed in the revised version of our manuscript.

1. *The assertion that the 'dark' function of CPDphr is adaptive in cave environments (lines 102-103 and 405-408) and that selection to maintain a functional copy resulted in the high sequence conservation compared to other UV light damage response genes would benefit from additional evidence.*

Please provide information on the colony (how many generations in captivity, how many breeding individuals). This is important for putting the genetic variation into context. Could the conservation/low sequence diversity be related to chance of collecting and then breeding individuals that randomly had little genetic diversity in that genomic region in the first place? Is the difference in sequence diversity between CPDphr and 4-6PPs reflected in wild samples? When only comparing two genes, it's difficult to tell which gene may be driving the difference (e.g. is LOF of 6-4 PPs, high conservation of CPDphr, or both driving the differences). Additional explanations for the lack of diversity of CPDphr are not explored and limited population genetic data from the lab colony are not sufficient to say that selection is the cause of the observed lack of sequence diversity. For example, tight linkage to another gene under selection could also result in the observed lack of diversity in CPDphr or perhaps most of the genome has very low diversity and the LOF alleles and relaxed purifying selection result in high sequence diversity. Context about whether the genes are of similar length or comparison to other genes could be a simple way to boost support for this observation.

This is an excellent point. The reviewer's comments are correct and give us the opportunity to clarify some issues that, mistakenly, we took for granted in the previous version.

In a new supplementary Table 1, we have now reported from which generation each of the individual adult fish were derived that we sequenced for Figure 1. Specifically, two samples (SeqID 6 and 7) originated from the F1 population born from wild specimen parents. The remaining samples were taken from F2 individuals produced from the F1 population that originated from the specimens collected in Somalia in 1982. In this regard, it is important to note that in common with many cave species, *P. andruzzii* has a long generation time: it is particularly long-lived (some specimens in the colony are at least 40 years old) and also sexual maturity is only reached after 10 years. Therefore, the colony of about 100 individuals consists mainly of F0, F1 and F2 generation individuals.

More than 100 breeding specimens of *Phreatichthys andruzzii* collected in 1982 from the wild at Bud Bud, Somalia (Latitude 04°11'19"N; Longitude 46°28'27"E; at an altitude of 137 m) were originally used to establish our laboratory colony and were expected to represent the original population in terms of genetic variation. Unfortunately, due to the dangerous geopolitical situation

in Somalia, it is currently not possible for us to collect wild fish and so we simply cannot assess the sequence diversity of the photolyase genes in wild samples today (<https://sonna.so/en/somali-national-army-captures-bud-bud-area-under-galmudug-state/>). However, there are various clues as to factors which may have influenced the genetic diversity that is evident in our laboratory population.

Following desertification, the Somalian cavefish population remained isolated in complex, water-filled subterranean phreatic layers probably experiencing a reduction in effective population size compared to the ancestral surface population. The level of gene flow between the various phreatic layers that have been colonized by the fish, if any, is not known. Two papers investigated the genetic variability of the sampled populations and obtained slightly different results: Cobolli Sbordoni et al (1996) using allozyme markers found a relatively low level of heterozygosity within and between populations collected from different wells in central Somalia and concluded that this could result from the isolation typical of subterranean populations¹. However, Colli et al. (2009) found a higher genetic variation at the mitochondrial DNA level and did not confirm the population structure predicted using nuclear DNA markers².

The reduction in effective population size due to the groundwater colonization may have affected the genomic variation, so reducing the level of polymorphism. However, several lines of evidence suggest that this is not the most parsimonious explanation for the complete lack of variation of CPDpfr in *P. andruzzii*: (i) Our results show an extremely conserved gene, with no variation in 14 chromosomes across 14kb including synonymous sites and intronic sequences; (ii) The 6-4pfr sequences are 1/3 in length (about 5 kb) and carry 8 polymorphic sites; (iii) Our previous work has shown that, similarly to what we observe here for the 6-4pfr gene, other genes involved in light-related pathways carry loss of function mutations and have drifted towards partial or complete pseudogenization in *P. andruzzii*. Specifically, the melanopsin gene *opn4m2*, TMT opsin and the rhodopsin visual photoreceptor gene in *P. andruzzii* exhibit multiple loss-of-function mutations, while for other opsin genes we found no evidence for any loss-of-function (Calderoni et al. 2016, Cavallari et al., 2011 and our unpublished data)^{3,4}. Furthermore, while most core circadian clock genes are highly conserved in *P. andruzzii*, the light-inducible period 2 clock gene carries mutations which result in aberrant splicing and consequently an abnormal, C-terminally truncated, cytoplasmic protein^{4,5}. From a broader evolutionary perspective, it is well established that both CPDpfr and 6-4pfr are highly conserved genes within the photolyase/cryptochrome superfamily, maintained by strong purifying selection due to their essential role in DNA repair^{6,7}. However, their conservation is not uniform across all taxa, as gene loss and functional divergence have occurred in specific evolutionary lineages.

Considering all these factors and by using the approach reported in Calderoni et al. (2016)³, we have interpreted the 6-4pfr pseudogenization as being the consequence of relaxed natural selection. This relaxation likely results from regressive evolution due to the complete loss of its primary selective agents, namely UV radiation and visible light. Within this framework, we propose that the extreme level of conservation of CDPpfr suggests the maintenance of a constraint on this gene, potentially indicating an alternative functional role. Nevertheless, alternative explanations cannot be completely ruled out, and so we now discuss these in our revised Discussion section.

2. The use of barplots is not ideal as they obscure the distribution of the actual data. <https://www.nature.com/articles/nmeth.2807> Please use data visualization (e.g. box plots, violin plots) that can better represent the variation.

We thank the reviewer for raising this important point. We have now replotted the bar graphs originally present in Figures 2(a, b, c and d), 3(a), 4(a, b, c, d, e, f, h and i), 5(a) and Supplementary Figure 8(a). In addition, for these sets of results we have now combined all the available biological replicates so that the plot provides an accurate view of the variation observed in all replicate experiments performed. Consequently, our results more convincingly support our original conclusions.

3. *This work applies many techniques, and a broad audience may find it hard to keep track of the impressive set of assays use help test the central hypothesis that CPDpnr functions in the dark and has a large role repairing DNA damage beyond UV damage.*

*One option to add clarity could be that when a specific cell line vs whole tissue is introduced, rationale for why using that cell line or tissue is included (e.g. lines 85-86,145-148, 216-218). Authors do a nice job introducing mammalian cell line, but the medaka vs zebrafish, cell vs fin clip could also use similar treatment. Such additions would help readers follow along with the analyses and provide justification for using such cell lines from organisms that aren't the focal cave fish *P. andruzzii*, while highlighting the extensive capacity for 'dark' CPDpnr activity across diverse organisms/cellular contexts.*

The Reviewer makes a valuable suggestion, and we have now adapted the text so that each time we introduce the use of a new model, we explain more clearly the rationale behind this particular step.

Minor comments

1. L60. *Please avoid the term "higher" vertebrates, it implies a hierarchy in evolution that doesn't scientifically exist.*

Absolutely! We totally agree!! We have now corrected this point and apologise for our error of including this erroneous terminology.

2. *Lines 178-206 CPDpnr transfected 3T3 vs non-transfected 3T3. I am not experienced with many of the assays done in this study, but is the most appropriate control in this comparison supposed to be between two transfected cell lines (one with the gene of interest, one with a sort of control gene not expected to have a similar function) to rule out a role of the transfection process in the cells ability to handle stressful conditions that follow? If that's not a standard control in these types of assays then perhaps one additional sentence about how the transfection process isn't expected to skew the results would help a broad audience better understand the assay (or if it is a more ideal control but practicality/resources didn't allow, then a caveat sentence iwould be warranted).*

The Reviewer is of course right to express caution over whether our choice of controls for the 3T3 transfection experiments is appropriate. That our 3T3 results are not the consequence of the transfection process is confirmed by the data we present in Figure 5 panel g and Supplementary Figure 10 panels b and c. Here are results from 3T3 cells stably transfected with expression vectors encoding single amino acid mutated versions of CPDpnr which abolish the photoreactivation repair function of the protein and these mutant-expressing cell lines behave in an identical fashion to the untransfected 3T3 cells. We have now added this information to the legends for Figure 3 and Supplementary Figure 4.

3. *Lines 216-223: In the comparison of iCab medaka to CPDpnr medaka(?) fish – is the only big difference expected between these two lines the knockdown mutation of CPDpnr? Some*

explanation about the differences between these two lines should be mentioned to help readers interpret the resulting differences in CPD levels being solely due to having a functional CPDphr or not.

This is an important point. One of the major differences between medaka and zebrafish is that medaka is able to tolerate extensive inbreeding. As a result, wildtype medaka lines such as iCab have experienced close to 100 generations of brother-sister crosses and so can be considered effectively isogenic. Despite many attempts, only a limited number of generations of inbreeding for zebrafish has been possible, ultimately resulting in single gender families or sterile offspring. As a result, the commonly used wildtype lines of zebrafish are typically far more polymorphic and so the origin of phenotypes in mutants can be more problematic to explain. Instead, the photolyase mutant lines of medaka as well as the wild type lines all share the same iCab isogenic background and so we can confidently predict that phenotypic differences observed in these mutants specifically relate to the photolyase mutations and are not the result of linkage with other polymorphic genes. We have now explained this important point in our results section when we first introduce the use of these medaka lines.

4. Figure 2e: From eyeballing the counts across the two panels, it appears that far fewer cells (dead or alive) were accounted for in right panel compared to the left panel despite even numbers of cells going into the cell viability assay being mentioned in the methods. Is this an additional sign that the cells without functional CPDphr were so damaged/dead from the H2O2 exposure that they couldn't be accounted for during the counting process? Or something else? Could be helpful to explain this pattern when talking about lower alive cell counts in the results section.

Thank you for pointing this out, as it may also be confusing to the reader. In fact, our assay determines not only cell viability but also total cell number. Cell counts are affected by a) reduced proliferation and b) by severe cell death which results in loss of cell counts due to cellular disintegration and consequent detachment from the culture surface. We have therefore amended our statements in the corresponding legend of figure 2 accordingly.

5. Supplementary Figures 3, 5, 6, 8, 7, 9. Can you please add a biological interpretation to the legend for the reader (i.e. what is the take home message from this?)

The Reviewer is correct to point out a potential lack of clarity for the reader in terms of the meaning of these figures. To remedy this problem, we have now adapted the Figure legends text, in particular including short titles, in order to clearly spell out the take-home message for each figure.

6. Figure S4. How do you interpret the more cell viability in LD than DD? Can you explain a bit more for the reader?

Again, this Supplementary Figure suffers a little from an incomplete explanation in the Figure legend, of the meaning of the presented data. We have now expanded the text to explain that these experiments constitute a control for the function of the ectopically expressed CPD photolyase in conferring enhanced DNA repair and cell survival following UV damage upon exposure to light-dark cycle but not constant dark conditions.

7. Figure 2. Please specify whether these were LD or DD conditions.

All experiments presented in Figure 2 were performed in DD conditions, an important point that is now stated clearly in the legend for this figure. More generally, additional experimental details including the duration of treatments and recovery periods are now provided in the Material and Methods section.

8. *Figure S8. Is there a control for LD? It doesn't appear that the immunofluorescence in the panel (b) for 36hrs and 48hrs matches what is shown in the bargraph for (a). Perhaps more explanation is needed?*

We agree with the Reviewer that in the cell images we selected, the characteristic fine punctate nuclear staining obtained with the H2AX antibody were not easily visible in these medaka cells. We have now enlarged these images so that the staining can be seen more easily.

Reviewer 3

We thank this reviewer for their very positive comments. They felt that „...*The manuscript is well written, the statistics are appropriate, and the findings are important as they challenge the long-standing view that photolyases are exclusively light-dependent enzymes.*“ Furthermore, they concluded that „*This work has broad implications for evolutionary biology, DNA repair research, and even biomedical applications related to oxidative stress-induced damage*“.

They made several valid points that we have now addressed in the revised version of our manuscript:

1. *Tissue-specific discrepancy: The study reports that CPDphr prevents CPD accumulation in the liver but not the brain while protecting against 8-OHdG in both. The reason for this discrepancy is presented but not fully explored. Does the brain have alternative repair mechanisms? Is the liver more susceptible to ROS-induced CPD formation? Addressing this gap would provide a clearer picture of how CPDphr is selectively active across tissues.*

This is an excellent question. Many lines of evidence have pointed to cell type-specific differences in the relative activities of DNA repair systems⁸⁻¹⁰. In addition, the response to DNA damage differs significantly over the course of the cell cycle as well as in tissue-specific stem cells^{11,12}. Furthermore, fundamental differences in the oxidative stress response of hepatocytes and neurons may account for differences in DNA repair systems as well as different types of DNA damage accumulating in these two tissues^{13,14}. These different properties may well ultimately result in differential accumulation of DNA damage in the tissues we tested. We now raise this important issue in our revised Discussion section.

2. *3T3 cells are useful tools for studying gain of function, though it's not clear why this line was chosen. Might a better line be one that completely lacks all UV-damage repair pathways (XP-A fibroblasts). This could be used to determine whether the observed protection is independent of existing repair mechanisms.*

While it would certainly be interesting to test gain of CPDphr function in XP-A fibroblasts, this would represent the first step of a major study to explore precisely which DNA repair systems accommodate the CPD photolyase dark function. Our decision to employ 3T3 cells was based more simply on this being an easily transfectable cell line which, since it is derived from mouse, is well documented to lack photolyases and photoreactivation DNA repair. Therefore, they represent ideal cell models to explore the functional consequences of overexpression of a photolyase in a photoreactivation repair deficient background. However, we do certainly agree

that the next logical steps should certainly be to identify the DNA repair systems that cooperate with CPDphr, a point we now refer to in the last part of our revised Discussion section.

3. *The study attributes the protective effect of CPDphr primarily to its direct role in DNA repair, specifically in repairing CPD and 8-OHdG lesions under oxidative stress. However, there are several alternative mechanisms that could contribute to the observed protective effects. Specifically, CPD-phr could interact with oxidative stress response pathways or be broadly involved in modulating redox homeostasis. It's unclear to me how a direct link to DNA damage repair complex is deduced as presented in Fig 6.*

Key experiments that link CPDphr with a potentially direct role in DNA damage repair are our electrophoretic mobility shift assays which demonstrate the ability of CPDphr to interact directly with 8-OHdG lesion containing DNA. Furthermore, CPDphr has of course been exhaustively characterised as a protein that can interact with (and repair) DNA damage. Given these results, we felt that exploring what could potentially be more indirect links between CPDphr and redox homeostasis would constitute a major experimental effort that would be somewhat tangential to the rational direction of the current study. For all these reasons, we chose to focus this work on exploring CPDphr functionality at the level of DNA repair. Nevertheless, at this stage of course we cannot exclude potential indirect links between CPDphr function and intracellular levels of ROS, and so we have now acknowledged this point in the revised Discussion section. Also, we have added a dotted arrow in our summary Figure 6 between the DNA repair machinery and CPDphr to illustrate that this is one (the most likely direct) link but that theoretically there may be other indirect connections.

Minor points

The manuscript is very well written, but a few grammatical errors can be cleaned up:

Line 368–370: “Based on our results, we propose the following model for DNA damage repair involving dual functionality of CPDphr (Fig. 6).” It took me a while to understand what this sentence said. I'd suggest something simpler like “involving two distinct functions of CPDphr”

Line 568–570: “Fin clips were recovered with a sterile pipette tip and transfer into a tube filled with absolute ethanol.” Should be “and transferred.”

Line 570–572: “The permission to perform experiments with cavefish was approved by Italian Ministry of Health (aut. N. 890/2016-PR).” Should be “by the Italian Ministry of Health”

We absolutely agree! These errors have now been corrected, and we have also carefully proofread the revised version text to ensure that any similar grammatical errors have been avoided.

References

1. Sbordoni, M. C., Matthaeis, E., Mattocchia, M., Berti, R. & Sbordoni, V. Genetic variability and differentiation of hypogean Cyprinid fishes from Somalia. *Journal of Zoological Systematics and Evolutionary Research* 34, 75–84 (1996).
2. Colli, L., Paglianti, A., Berti, R., Gandolfi, G. & Tagliavini, J. Molecular phylogeny of the blind cavefish *Phreatichthys andruzzii* and *Garra barreimiae* within the family Cyprinidae. *Environ Biol Fishes* 84, 95–107 (2009).

3. Calderoni, L. *et al.* Relaxed selective constraints drove functional modifications in peripheral photoreception of the cavefish *P. andruzzii* and provide insight into the time of cave colonization. *Heredity (Edinb)* 117, 383–392 (2016).
4. Cavallari, N. *et al.* A blind circadian clock in cavefish reveals that opsins mediate peripheral clock photoreception. *PLoS Biol* 9, (2011).
5. Ceinos, R. M. *et al.* Mutations in blind cavefish target the light-regulated circadian clock gene, period 2. *Sci Rep* 8, 8754 (2018).
6. Lucas-Lledo, J. I. & Lynch, M. Evolution of Mutation Rates: Phylogenomic Analysis of the Photolyase/Cryptochrome Family. *Mol Biol Evol* 26, 1143–1153 (2009).
7. Mei, Q. & Dvornyk, V. Evolutionary History of the Photolyase/Cryptochrome Superfamily in Eukaryotes. *PLoS One* 10, e0135940 (2015).
8. Sabatella, M., Thijssen, K. L., Davó-Martínez, C., Vermeulen, W. & Lans, H. Tissue-Specific DNA Repair Activity of ERCC-1/XPF-1. *Cell Rep* 34, 108608 (2021).
9. Hoch, N. C. Tissue Specificity of DNA Damage and Repair. *Physiology* 38, 231–241 (2023).
10. Dion, V. Tissue specificity in DNA repair: lessons from trinucleotide repeat instability. *Trends in Genetics* 30, 220–229 (2014).
11. Clay, D. E. & Fox, D. T. DNA Damage Responses during the Cell Cycle: Insights from Model Organisms and Beyond. *Genes (Basel)* 12, 1882 (2021).
12. Blanpain, C., Mohrin, M., Sotiropoulou, P. A. & Passegué, E. DNA-Damage Response in Tissue-Specific and Cancer Stem Cells. *Cell Stem Cell* 8, 16–29 (2011).
13. Kim, J. M., Kim, H. G. & Son, C. G. Tissue-Specific Profiling of Oxidative Stress-Associated Transcriptome in a Healthy Mouse Model. *Int J Mol Sci* 19, 3174 (2018).
14. Cucchi, D., Gibson, A. & Martin, S. a. The emerging relationship between metabolism and DNA repair. *Cell Cycle* 20, 943–959 (2021).